# Relevance of Preoperative Hyperbilirubinemia in Patients Undergoing Hepatobiliary Resection for Hilar Cholangiocarcinoma

**DOI:** 10.3390/jcm8040458

**Published:** 2019-04-05

**Authors:** Karolina Maria Wronka, Michał Grąt, Jan Stypułkowski, Emil Bik, Waldemar Patkowski, Marek Krawczyk, Krzysztof Zieniewicz

**Affiliations:** 1Liver and Internal Medicine Unit, Department of General, Transplant and Liver Surgery, Medical University of Warsaw, 1A Banacha Street, 02-097 Warsaw, Poland; 2Department of General, Transplant and Liver Surgery, Medical University of Warsaw, 1A Banacha Street, 02-097 Warsaw, Poland; michal.grat@gmail.com (M.G.); jan.styp@gmail.com (J.S.); biku94@wp.pl (E.B.); wpatek@mp.pl (W.P.); marek.krawczyk@wum.edu.pl (M.K.); krzysztof.zieniewicz@wum.edu.pl (K.Z.)

**Keywords:** hilar cholangiocarcinoma, bilirubin, biliary drainage, mortality, morbidity, prediction

## Abstract

Preoperative hyperbilirubinemia is known to increase the risk of mortality and morbidity in patients undergoing resection for hilar cholangiocarcinoma. The aim of this study was to characterize the associations between the preoperative bilirubin concentration and the risk of postoperative mortality and severe complications to guide decision-making regarding preoperative biliary drainage. Eighty-one patients undergoing liver and bile duct resection for hilar cholangiocarcinoma between 2005 and 2015 were analyzed retrospectively. Postoperative mortality and severe complications, defined as a Clavien–Dindo grade of ≥III, were the primary and secondary outcome measures, respectively. The severe postoperative complications and mortality rates were 28.4% (23/81) and 11.1% (9/81), respectively. Patients with preoperative biliary drainage had significantly lower bilirubin concentrations (*p* = 0.028) than did those without. The preoperative bilirubin concentration was a risk factor of postoperative mortality (*p* = 0.003), with an optimal cut-off of 6.20 mg/dL (c-statistic = 0.829). The preoperative bilirubin concentration was a risk factor of severe morbidity (*p* = 0.018), with an optimal cut-off of 2.48 mg/dL (c-statistic = 0.662). These results indicate that preoperative hyperbilirubinemia is a major risk factor of negative early postoperative outcomes of patients who undergo surgical treatment for hilar cholangiocarcinoma and may aid in decision-making with respect to preoperative biliary drainage.

## 1. Introduction

Hilar cholangiocarcinoma is the most prevalent type of malignancy and accounts for 50–60% of all malignancies in the bile ducts [1,2,3]. Extrahepatic bile duct resection, with or without major hepatic resection, with negative surgical margins and potential hepatoduodenal lymphadenectomy remains the gold-standard treatment of patients with hilar cholangiocarcinoma, which has a low response to chemotherapy and radiation therapy [2,3,4]. However, the surgical treatment of hilar cholangiocarcinoma is associated with a relatively high risk of postoperative morbidity and mortality [5]. Preoperative hyperbilirubinemia that is associated with a bile duct stricture appears to be the most important and modifiable risk factor of postoperative mortality and complications [5]. An elevated bilirubin concentration is associated with a higher risk of postoperative liver failure, intraoperative blood loss, postoperative bleeding, infection, heart failure, renal insufficiency, and death in the early postoperative period than a lower bilirubin concentration [5,6,7,8,9,10].

However, the preoperative bilirubin concentration associated with an unacceptable risk of postoperative mortality and severe complications, and the indication for postponing surgery and performing preoperative biliary drainage has not been established [4,5,11]. Preoperative biliary drainage is recommended in patients with malnutrition, hypoalbuminemia, cholangitis, liver failure, or radiochemotherapy, or those who are selected to undergo portal vein embolization [7,12]. This procedure is associated with the risk of iatrogenic cholangitis, postoperative bacteremia, wound infection, longer hospitalization duration, and higher treatment costs [1,2,8,11,13,14].

The association between the preoperative bilirubin concentration and the risk of postoperative mortality and severe complications has not been precisely established yet. The purpose of this study was to precisely characterize the associations between the preoperative serum bilirubin concentration and the risk of postoperative mortality, as well as severe postoperative complications, to propose guidelines for preoperative biliary drainage that would aid decision-making processes.

## 2. Materials and Methods

The study protocol was approved by the institutional review board of the Medical University of Warsaw.

In this retrospective, observational study, we evaluated the data of 81 patients who underwent liver and bile duct resection for hilar cholangiocarcinoma at the Department of General, Transplant, and Liver Surgery at the Medical University of Warsaw, Poland, between January 2005 and December 2015. Patients with grade I (Bismuth–Corlette) tumors and those who underwent only extrahepatic bile duct resection were excluded from further investigation. 

The primary and secondary outcome measures were postoperative in-hospital mortality and the occurrence of severe postoperative complications that were defined as a Clavien–Dindo grade III or higher, respectively [15]. The preoperative serum bilirubin concentration was the primary factor of interest. Patients were divided into two groups regarding whether or not they underwent preoperative biliary drainage. 

The data are presented as numbers with percentages (categorical data) and medians with interquartile ranges (continuous data). Variables were compared using Mann–Whitney’s U, χ^2^, and Fisher’s exact tests, wherever appropriate, between patients with preoperative biliary drainage and those without, as well as between patients who underwent right or extended right hemihepatectomy and those who underwent left hemihepatectomy. A logistic regression analysis was used to evaluate the risk factors of the established endpoints. Variables were included in multivariable models based on the results of the univariable analyses. The multivariable analysis was conducted using the backward elimination method. The relationship between the preoperative bilirubin concentration and the established endpoints was assessed in subgroups of patients with and without preoperative biliary drainage and patients who underwent right or extended right hemihepatectomy and those with left hemihepatectomy, and subgroups based on the median of other continuous risk factors.

Receiver operating characteristic curves were constructed to discover the optimal cut-off values for the bilirubin concentration to predict the established outcomes in the whole group and those among the subgroups in which the preoperative bilirubin concentration was significantly associated with the analyzed endpoints. Odds ratios (ORs) and areas under the curve (c-statistics) are presented with 95% confidence intervals (95% CIs). The level of statistical significance was set at 0.05. All statistical analyses were performed using Statistica 12 software (Dell Inc., Tulsa, OK, USA). 

## 3. Results

The baseline characteristics of the patients in this study and comparisons between those with (*n* = 58, 71.6%) and those without (*n* = 23, 28.4%) preoperative biliary drainage are presented in Table 1. Regarding the method of preoperative biliary drainage, endoscopic stenting was performed in 57 patients (98.3%) and percutaneous transhepatic drainage was performed in 1 (1.7%). Patients with preoperative biliary drainage had a lower preoperative bilirubin concentration (*p* = 0.028), higher rate of positive bile cultures (*p* < 0.001), higher gamma-glutamyl transpeptidase activity (*p* = 0.006), higher rate of right or extended right hemihepatectomy (*p* = 0.003), and shorter operative time (*p* = 0.025) than did those without biliary drainage. Patients undergoing right or extended right hemihepatectomy had a higher rate of preoperative biliary drainage (*p* = 0.003), higher gamma-glutamyl transpeptidase activity (*p* = 0.048), and lower albumin concentration (*p* = 0.014) than did those who underwent left hemihepatectomy. 

Overall, the postoperative mortality rate was 11.1% (nine patients); the causes of death were postoperative failure of a liver remnant (*n* = 7), hemorrhage (*n* = 1), and multiple-organ failure because of aspiration (*n* = 1). The postoperative mortality rate was 13.8% (8/58) in patients who underwent preoperative biliary drainage and 4.4% (1/23) in patients without preoperative biliary drainage (*p* = 0.434). The postoperative mortality rate was significantly higher in patients who underwent right or extended right hemihepatectomy (22.2%; 8/36) than in those who underwent left hemihepatectomy (2.2%; 1/45; *p* = 0.009). Twenty-three patients (28.4%) developed severe postoperative complications (Table 2). In particular, severe complications were observed in 29.3% (17/58) of patients who underwent preoperative biliary drainage and 26.1% (6/23) of patients without preoperative biliary drainage (*p* = 0.999). The severe complications rates of patients who underwent right or extended right hemihepatectomy and those who underwent left hemihepatectomy were 36.1% (13/36) and 22.2% (10/45), respectively (*p* = 0.217).

The preoperative bilirubin concentration (*p* = 0.003) was significantly associated with postoperative mortality. The other risk factors of postoperative mortality were the preoperative hemoglobin concentration (*p* < 0.001), preoperative albumin concentration (*p* = 0.003), intraoperative blood transfusions (*p* = 0.013), and right or extended right hemihepatectomy (*p* = 0.020; Table 3). The optimal bilirubin concentration cut-off value to predict postoperative mortality in all patients was ≥6.20 mg/dL (c-statistic = 0.829; 95% CI: 0.723–0.934), with a sensitivity of 88.9%, specificity of 73.2%, negative predictive value of 98.1%, and positive predictive value of 29.6% (Figure 1). In subgroup analyses, the preoperative bilirubin concentration was a significant risk factor of postoperative mortality in patients with a preoperative albumin concentration that was less than or equal to the median (3.6 g/dL; *p* = 0.019) or preoperative hemoglobin concentration that was less than or equal to the median (12.45 g/dL; *p* = 0.014), as well as patients who underwent preoperative biliary drainage (*p* = 0.002) and those who underwent right or extended right hemihepatectomy (*p* = 0.004; Table 4). The bilirubin concentration cut-off values to predict postoperative mortality (≥6.20 mg/dL) were identical among patients with a preoperative albumin concentration ≤3.6 g/dL (c-statistic = 0.788; 95% CI: 0.624–0.952), preoperative hemoglobin concentration ≤12.45 g/dL (c-statistic = 0.797; 95% CI: 0.650–0.945), and those who underwent preoperative biliary drainage (c-statistic = 0.891; 95% CI: 0.800–0.982), except for patients who underwent right or extended right hemihepatectomy, in whom the bilirubin concentration cut-off value was ≥2.73 mg/dL (c-statistic = 0.933; 95% CI: 0.853–1.000; Table 5).

The preoperative bilirubin concentration was significantly associated with severe morbidity (*p* = 0.018), as were the creatinine (*p* = 0.003), urea (*p* = 0.028), and hemoglobin concentrations (*p* = 0.019) (Table 3). The optimal bilirubin concentration cut-off value to predict the occurrence of severe postoperative complications was ≥2.48 mg/dL (c-statistic = 0.662; 95% CI: 0.526–0.799), with a sensitivity of 73.9%, specificity of 59.6%, negative predictive value of 85.0%, and positive predictive value of 42.5% (Figure 2). Moreover, the preoperative bilirubin concentration significantly influenced the patient’s risk of developing severe postoperative complications among those with a preoperative creatinine concentration that was greater than or equal to the median (0.77 mg/dL; *p* = 0.032), preoperative urea concentration that was greater than or equal to the median (28 mg/dL; *p* = 0.019), patients who underwent preoperative biliary drainage (*p* = 0.042), and patients who underwent right or extended right hemihepatectomy (*p* = 0.020; Table 4). The optimal bilirubin concentration cut-off values to predict severe complications were ≥6.20 mg/dL in patients with a preoperative creatinine concentration ≥0.77 mg/dL (c-statistic = 0.724; 95% CI: 0.559–0.888) and preoperative urea concentration ≥28 mg/dL (c-statistic = 0.741; 95% CI: 0.576–0.906) and ≥2.48 mg/dL in patients who underwent preoperative biliary drainage (c-statistic = 0.685; 95% CI: 0.530–0.840) and those who underwent right or extended right hemihepatectomy (c-statistic = 0.764; 95% CI: 0.585–0.944; Table 5).

According to the multivariable analysis, the preoperative creatinine (*p* < 0.001, OR = 9.10; 95% CI: 2.71–30.53) and hemoglobin concentrations (*p* = 0.009, OR = 0.59; 95% CI: 0.39–0.87) were independent risk factors of severe complications.

## 4. Discussion

The present study confirms that preoperative hyperbilirubinemia is strongly associated with postoperative mortality and morbidity in patients who undergo surgery for hilar cholangiocarcinoma, especially in those with initially high risk. Based on the obtained results, recommendations for preoperative biliary drainage in patients with hilar cholangiocarcinoma who are qualified to undergo liver and bile duct resection can be made. For clinical use, two cut-off points of the preoperative bilirubin level were established: ≥6.00 mg/dL and ≥2.50 mg/dL. The outcomes of this study indicate that preoperative biliary drainage should be performed in all patients with hilar cholangiocarcinoma with a bilirubin concentration ≥ 6.00 mg/dL, whereas it should be considered in patients with a bilirubin concentration <6.00 mg/dL and ≥2.50 mg/dL, especially in patients with preoperative hypoalbuminemia, anemia, or renal dysfunction, those who are scheduled to undergo right or extended right hemihepatectomy, and those with ineffective biliary drainage. Preoperative biliary drainage is not recommended in patients with a preoperative bilirubin concentration <2.50 mg/dL.

The relationship between preoperative hyperbilirubinemia and the risk of postoperative mortality was strongly confirmed in prior reports. In particular, Jarnagin et al. proved that preoperative hyperbilirubinemia was risk factor of postoperative death in patients undergoing liver resection [16]. Similarly, Belghiti et al. noticed that the mortality rate among patients with preoperative hyperbilirubinemia was significantly higher than that among other patients [17]. 

In the present study, the optimal bilirubin concentration cut-off to predict postoperative mortality was ≥6.20 mg/dL with a positive predictive value of 29.6% and high sensitivity level of 87.5%, which are already valuable parameters to predict postoperative mortality. The results that were obtained from high-risk patients were even more satisfying. The positive predictive values for the same bilirubin concentration among patients with hypoalbuminemia and anemia and those who underwent preoperative biliary drainage were higher, at 41.2%, 47.1%, and 50.0%, respectively, which indicates that these patients were more likely to die in the early postoperative period than those with lower values. The optimal bilirubin concentration cut-off value to predict postoperative mortality was 2.73 mg/dL in patients who underwent right or extended right hemihepatectomy, with a sensitivity of 100.0% and positive predictive value of 57.1%, which is higher than the value in the whole group, even though the bilirubin concentration was more than twice lower. 

Regarding severe postoperative complications, preoperative hyperbilirubinemia was also a significant risk factor, which was widely described in previously published studies [8,18,19]. Miyagawa et al. observed that patients who developed postoperative complications were characterized with a significantly higher preoperative bilirubin concentration than did those without complications [18]. According to Ribero et al.’s research, a preoperative bilirubin concentration >3 mg/dL was an independent risk factor of postoperative liver failure [19], which was a prevalent complication in the studied group. In the present analysis, the optimal bilirubin concentration cut-off to predict severe postoperative complications was ≥2.48 mg/dL in the whole group, with a positive predictive value of 42.5% and sensitivity of 73.9%. The same bilirubin concentration was the optimal cut-off among patients who underwent preoperative biliary drainage and those who required right or extended right hemihepatectomy, with higher positive predictive values of 48.0% and 66.7%, respectively. A higher optimal bilirubin concentration cut-off value of ≥6.20 mg/dL was associated with a higher positive predictive value of approximately 70% in patients with renal dysfunction. Preoperative hyperbilirubinemia led to severe postoperative complications in patients with renal dysfunction, those who underwent ineffective preoperative biliary drainage, and those requiring right or extended right hemihepatectomy. Although the multivariable analysis did not prove that there was a significant association between preoperative hyperbilirubinemia and severe postoperative complications, we do not deny its significant influence, as this could be caused by the limited number of patients in this study and strong influence of renal dysfunction.

The role of preoperative biliary drainage in patients with obstructive jaundice remains controversial [20,21]. However, preoperative biliary tree decompression is recommended in some patients regarding their general condition and comorbidities such as malnutrition, hypoalbuminemia, biliary sepsis, cholangitis, liver insufficiency, and renal failure, as well as in those with preoperative portal vein embolization and chemoradiation therapy [7,12]. Considering the lack of guidelines stating the precise preoperative bilirubin concentration associated with an unacceptably high risk of postoperative mortality and severe complications, as well as the outcomes of the present study, we created the above-named proposal for recommendations for preoperative biliary drainage. 

A preoperative bilirubin concentration that indicates the need for preoperative biliary drainage varies considerably among published studies, and this concentration’s cut-off has not been analyzed regarding the risk of postoperative mortality and morbidity. Cai et al. concluded that a preoperative bilirubin concentration of 12.40 mg/dL indicated the need for preoperative biliary drainage [22], whereas some researchers claimed that a hyperbilirubinemia level of 10.00 mg/dL required preoperative biliary decompression [5,23]. However, in other reports, a preoperative bilirubin concentration >5.00 mg/dL was the reason for postponing surgery and decreasing the level of hyperbilirubinemia [20]. Additionally, Sano et al. considered a preoperative bilirubin concentration >3.00 mg/dL as an indication for preoperative biliary drainage [24]. Nakanishi et al. assessed an even lower preoperative bilirubin concentration value (>2.00 mg/dL) as an indication for biliary decompression [9].

Hyperbilirubinemia was an additional risk factor of postoperative mortality and morbidity in high-risk patients, and it could be strictly connected to poor outcomes. Hypoalbuminemia, which may coexist with obstructive jaundice because of malnutrition and intestinal absorptive disorders [25], was a significant risk factor of postoperative mortality. Furthermore, hypoalbuminemia is associated with the risk of postoperative liver failure, which may result in postoperative death [19] and was the most frequent reason for death in the analyzed data. In addition, preoperative anemia proved to be a risk factor of postoperative death and morbidity, which was widely observed in other studies [5,26,27,28]. This comorbidity may result in the need for intraoperative blood transfusions, which were also significantly associated with postoperative mortality in the present study. Similarly, Kennedy et al. found that intraoperative blood transfusions were a risk factor of postoperative liver failure and mortality following liver resection [29]. Moreover, renal insufficiency that was associated with increased preoperative creatinine and urea concentrations enhanced the risk of severe postoperative complications. Squires et al. analyzed data from three hospitals and concluded that patients with a creatinine concentration ≥1.8 mg/dL are at a risk of developing severe postoperative complications, including respiratory insufficiency and renal failure [30]. Finally, patients who undergo right or extended right hemihepatectomy had a higher mortality rate than did those who did not undergo this procedure, and they would probably benefit significantly from preoperative biliary decompression. In other studies, an increase in remnant liver failure because of jaundice was considered the main reason for death in the early postoperative period [7,19]. Other findings prove the necessity of decreasing the level of preoperative hyperbilirubinemia in patients who are qualified to undergo major liver resection [8,11,19,29]. Surprisingly, despite the strong influence of right or extended right hemihepatectomy on the risk of postoperative mortality, the relationship of this surgery with the risk of severe postoperative complications was not observed. The type of liver resection did not influence the occurrence of other severe complications, except for postoperative liver failure, which was the most prevalent reason for postoperative death. Considering the above correlation, patients who are scheduled for right or extended right hemihepatectomy would probably benefit from preoperative biliary drainage. Additionally, although preoperative biliary drainage was not associated with the risk of postoperative mortality and severe complications in the present analysis, the preoperative bilirubin concentration was a risk factor of both outcome measures among patients who underwent preoperative biliary drainage. Therefore, patients with ineffective preoperative biliary drainage would also benefit from decreasing the level of preoperative hyperbilirubinemia because of their probable poor condition and long-lasting hyperbilirubinemia. 

In the present study, preoperative biliary drainage was associated with a significantly high positive bile culture rate; therefore, patients who undergo preoperative biliary drainage may carry the risk of contamination. The relationship between preoperative biliary drainage and postoperative infectious complications was described in prior reports [6,21]. Significantly higher gamma-glutamyl transpeptidase activity in patients who underwent preoperative biliary drainage was possibly due to intervention within the biliary tree. Probably, there is no or little relevance of elevated gamma-glutamyl transpeptidase activity among patients with preoperative biliary drainage. However, preoperative bilirubin concentration was a risk factor of severe postoperative complications and mortality in this group of patients, therefore bilirubin concentration should be considered even among patients already with preoperative biliary drainage. Presumably because of selection bias, preoperative biliary drainage was performed more often in patients in worse general condition, mainly in patients requiring right or extended right hemihepatectomy, than in those with a better general condition. Furthermore, patients who underwent preoperative biliary drainage had a bilirubin level that was more than four times lower than that in patients without preoperative biliary drainage, which was indirect proof of the effectiveness of this procedure [6,21]. The presence of a stent in the biliary tree enables the surgeon to identify anatomical structures quickly, resulting in a shorter operation duration in patients who undergo preoperative biliary drainage.

TNM (tumor, nodes, metastasis) stage III or higher did not prove to be a significant risk factor for established endpoints. In the studied group, 14 patients with preoperative biliary drainage and 4 patients without preoperative biliary drainage were characterized with TNM tumor staging III or higher. The analysis included only patients qualified for potentially radical liver and bile duct resection. The small number of patients with advanced TNM stage may result from unresectability of advanced hilar cholangiocarcinoma. In numerous studies, the influence of high TNM stage on early outcomes is not investigated. At the same time, lower AJCC (American Joint Committee on Cancer) T stage has been shown to have positive impact on long-term outcome, including five-year survival [20,31] and disease-free survival [32]. However, long-term outcomes were not the subject of presented study.

Several limitations of the present study must be acknowledged. First, this was a retrospective study; therefore, it was characterized by all of the limitations that are associated with this type of research. Small number of patients included in the analysis may lead to exclusion of other significant relationships. Moreover, the duration of preoperative hyperbilirubinemia, as well as the duration and preferable method of effective preoperative biliary drainage, was not analyzed in the present study. In addition, we did not randomly select patients who were qualified for preoperative biliary drainage, and data about the indications for preoperative biliary drainage were not available; therefore, we did not assess the effectiveness of preoperative biliary drainage. Therefore, a prospective randomized study is warranted. 

## 5. Conclusions

In conclusion, the outcomes of this study indicate that preoperative hyperbilirubinemia is a major risk factor of poor early postoperative outcomes in patients who undergo surgical treatment of hilar cholangiocarcinoma and may aid in decision-making regarding preoperative biliary drainage. We established a cut-off value to indicate which patients should undergo this procedure. The results indicate that preoperative biliary drainage should be performed in all patients with hilar cholangiocarcinoma with a bilirubin concentration ≥6.00 mg/dL, whereas it should be considered in patients with a bilirubin concentration <6.00 mg/dL and ≥2.50 mg/dL, especially in patients with preoperative hypoalbuminemia, anemia, or renal dysfunction, those who are scheduled to undergo right or extended right hemihepatectomy, and those with ineffective biliary drainage. Preoperative biliary drainage is not recommended in patients with a preoperative bilirubin concentration <2.50 mg/dL. 

## Figures and Tables

**Figure 1 jcm-08-00458-f001:**
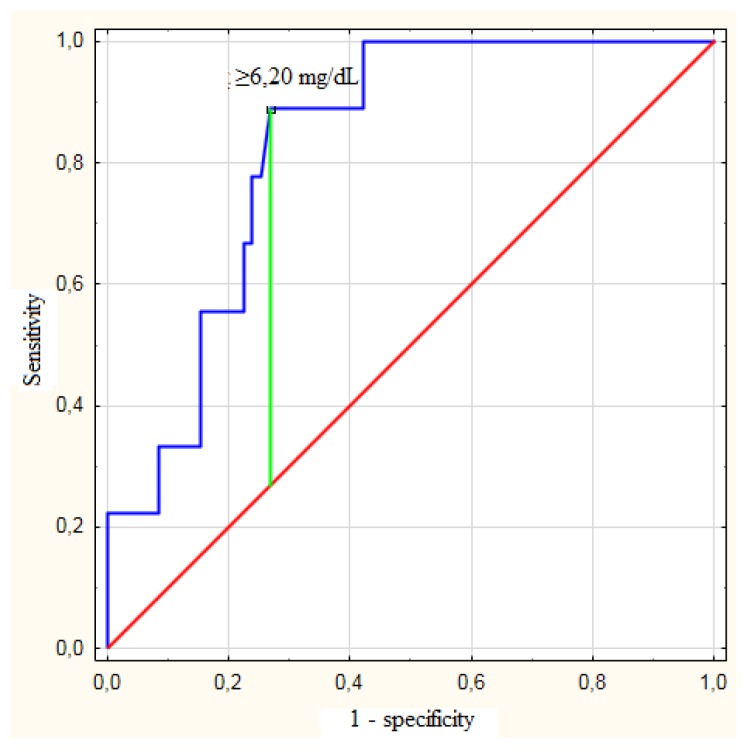
Receiver operating characteristics curves for prediction of postoperative mortality based on preoperative bilirubin concentration.

**Figure 2 jcm-08-00458-f002:**
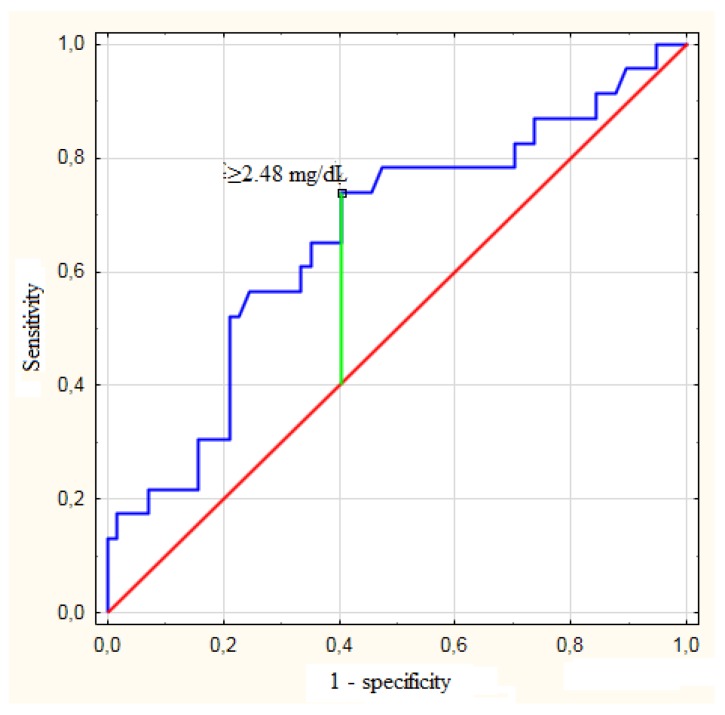
Receiver operating characteristics curves for prediction of occurrence of postoperative severe complications based on preoperative bilirubin concentration.

**Table 1 jcm-08-00458-t001:** Baseline characteristics of the study group and comparison between baseline characteristics of patients with and without preoperative biliary drainage.

Factors	Number with Percentage and Median with Interquartile Range (IQR)			
	Studied Group (*n* = 81)	Patients with Preoperative Biliary Drainage (*n* = 58)	Patients without Preoperative Biliary Drainage (*n* = 23)	*p*
Age	60 (54–65)	60.0 (55.0–67.0)	56.0 (48.0–65.0)	0.181
Patient sex				
Female	34 (42.0%)	24 (41.4%)	10 (43.5%)	0.999
Male	47 (58.0%)	34 (58.6%)	13 (56.5%)	
Bilirubin concentration (mg/dL)	2.44 (1.00–9.78)	2.14 (0.93–4.90)	9.64 (1.10–19.67)	0.028
AST activity (U/L)	66 (46–106)	72 (48–106)	59.5 (46–95)	0.406
ALT activity (U/L)	81 (51–106)	85 (51–110)	68.5 (58–85)	0.208
GGTP activity (U/L)	352 (156–592)	441 (210–643)	176 (105–298)	0.006
Creatinine concentration (mg/dL)	0.77 (0.70–0.91)	0.77 (0.69–0.88)	0.78 (0.72–1.07)	0.535
INR	1.00 (0.95–1.08)	0.99 (0.96–1.07)	1.03 (0.93–1.11)	0.562
Protein concentration (g/dL)	6.80 (6.35–7.35)	6.90 (6.40–7.50)	6.80 (6.30–7.00)	0.358
Albumin concentration (g/dL)	3.60 (3.20–3.90)	3.50 (3.05–3.90)	3.70 (3.50–3.80)	0.263
Urea concentration (mg/dL)	28.0 (22.0–36.0)	27.5 (21.0–36.5)	28.0 (22.0–32.0)	0.916
Hemoglobin concentration (g/dL)	12.45 (11.5–13.28)	12.3 (11.3–13.3)	12.7 (12.3–13.1)	0.124
White Blood Cells (10^3^/µL)	7.04 (5.90–8.59)	7.04 (6.30–8.41)	6.80 (5.55–9.00)	0.579
Platelets (10^3^/µL )	263.5 (213.0–326.0)	253.5 (206–349)	274 (227–308)	0.953
Ca19–9 (U/mL)	155.1 (43.1–421.6)	167.9 (53.4–562.0)	113.6 (26.9–222.0)	0.245
CEA (ng/mL)	1.95 (1.22–2.80)	2.0 (1.2–3.0)	1.8 (1.3–2.4)	0.678
TNM tumor staging III or higher	18 (22.2%)	14 (28.6%)	4 (21.1%)	0.760
Liver resection type				
Right or extended right	36 (44.4%)	32 (55.2%)	4 (17.4%)	0.003
Left	45 (55.6%)	26 (44.8%)	19 (82.6%)	
Packed red blood cells transfusion	0 (0–2)	0 (0–2)	2 (0–2)	0.384
Fresh frozen plasma transfusion	0 (0–2)	0 (0–2)	2 (0–3)	0.106
Operative time (hours)	7.0 (5.8–7.9)	6.5 (5.3–7.8)	7.5 (7.0–8.3)	0.025
Positive bile culture	46 (56.8%)	43 (74.1%)	3 (13.0%)	<0.001

Ca 19–9: Cancer antigen 19–9; CEA: Carcinoembryonic antygen, AST: Aspartate aminotransferase; ALT: Alanine aminotransferase; GGTP: Gamma-glutamyl transpeptidase; INR: International normalized ratio; TNM: tumor, nodes, metastasis.

**Table 2 jcm-08-00458-t002:** Severe postoperative complications in patients after liver and bile duct resection for hilar cholangiocarcinoma.

Postoperative Complications	Way of Treatment	Number of Patients with This Complication
Biliary leak	Endoscopic retrograde cholangiopancreatography	1
	Reoperation	1
	Percutaneous drainage	1
Bleeding	Reoperation	5
Rectus abdominal muscle bleeding	Wound revision	1
Wound infection, difficulty with drain removal	Minilaparotomy, wound revision	1
Abdominal abscess	Percutaneous drainage	1
	Reoperation	2
Intraabdominal infection	Reoperation	1
Wound dehiscence	Wound revision	3
Gastrointestinal tract bleeding	Gastroscopy	3
Pleural effusion	Pleurocentesis	3
Renal failure	Dialysis	3
Liver failure	Albumin dialysis	2
	Without albumin dialysis	6
Respiratory failure	Artificial ventilation	2
Heart failure	Pharmacological treatment	2
Multiple organ dysfunction syndrome as a result of choking	Pharmacological treatment, artificial ventilation	1
Death		9

**Table 3 jcm-08-00458-t003:** Risk factors of postoperative mortality and occurrence of severe complications after liver and bile duct resection for hilar cholangiocarcinoma.

Factors	Outcome Measure					
	Postoperative Mortality			Postoperative Severe Complications		
	OR	95% CI	*p*	OR	95% CI	*p*
Age	1.04	0.95–1.13	0.404	1.06	1.00–1.13	0.063
Patient sex (Male)	1.51	0.35–6.53	0.579	2.64	0.91–7.66	0.073
Preoperative biliary drainage	3.52	0.42–29.87	0.249	1.18	0.40–3.49	0.772
Bilirubin concentration (mg/dL)	1.14	1.05–1.24	0.003	1.08	1.01–1.15	0.018
AST activity (U/L)	1.97	0.63–6.19	0.245	0.96	0.35–2.62	0.941
ALT activity (U/L)	0.86	0.21–3.54	0.837	0.75	0.27–2.08	0.580
GGTP activity (U/L)	1.12	1.00–1.26	0.051	1.07	0.97–1.18	0.203
Creatinine concentration (mg/dL)	17.96	0.70–461.37	0.081	57.56	4.06–815.9	0.003
INR	9.33	0.08–1106.33	0.359	35.45	0.69–1827.7	0.076
Protein concentration (g/dL)	1.07	0.42–2.76	0.887	1.39	0.74–2.62	0.310
Albumin concentration (g/dL)	0.11	0.03–0.47	0.003	0.49	0.20–1.24	0.133
Urea concentration (mg/dL)	1.04	0.99–1.09	0.103	1.05	1.01–1,10	0.028
Hemoglobin concentration (g/dL)	0.27	0.13–0.56	<0.001	0.65	0.45–0.93	0.019
White Blood Cells (10^3^/µL)	1.02	0.72–1.43	0.922	1.04	0.82–1.32	0.770
Platelets (10^3^/µL )	0.91	0.45–1.85	0.790	0.84	0.51–1.39	0.501
TNM tumor staging III or higher	0.43	0.05–3.85	0.452	1.29	0.40–4.09	0.671
Right or extended right hemihepatectomy	12.6	1.49–106.02	0.020	1.98	0.74–5.26	0.172
Packed red blood cells transfusion	1.68	1.11–2.53	0.013	1.11	0.84–1.48	0.467
Fresh frozen plasma transfusion	1.34	0.99–1.82	0.057	1.08	0.84–1.38	0.549
Operative time (hours)	0.97	0.61–1.53	0.892	1.04	0.76–1.41	0.822
Positive bile culture	0.95	0.23–3.81	0.937	1.26	0.47–3.38	0.641

AST: Aspartate aminotransferase; ALT: Alanine aminotransferase; GGTP: Gamma-glutamyl transpeptidase; INR: International normalized ratio; TNM: tumor, nodes, metastasis. OR: Odds ratios; 95% CI: 95% confidence intervals. Odds ratios are given per: 1 mg/dL increase for bilirubin, 100 U/L increase for AST, ALT, and GGTP, 1 mg/dL increase for creatinine and urea, 1 increase for INR, 1 g/dL increase for protein, albumin, and hemoglobin, 10^3^/µL increase for white blood cells, 10^5^/µL increase for platelets, 1 unit increase for blood and plasma transfusions, 1 h increase for operative time.

**Table 4 jcm-08-00458-t004:** Preoperative bilirubin concentration as risk factor of postoperative mortality and occurrence of severe complications in high-risk groups of patients.

**Postoperative Mortality**			
**Group of Patients**	**OR**	**95% CI**	***p***
Patients with preoperative albumin concentration ≤3.6 g/dL	1.14	1.02–1.26	0.019
Patients with preoperative hemoglobin concentration ≤12.45 g/dL	1.13	1.03–1.25	0.014
Patients with preoperative biliary drainage	1.22	1.08-138	0.002
Patients after right or extended right hemihepatectomy	1.23	1.07–1.42	0.004
**Postoperative Severe Complications**			
**Group of Patients**	**OR**	**95% CI**	***p***
Patients with preoperative creatinine concentration ≥0.77 mg/dL	1.10	1.01–1.19	0.032
Patients with preoperative urea concentration ≥28 mg/dL	1.13	1.02–1.24	0.026
Patients with preoperative biliary drainage	1.10	1.00–1.21	0.042
Patients after right or extended right hemihepatectomy	1.15	1.02–1.29	0.020

OR: Odds ratios; 95% CI: 95% confidence intervals.

**Table 5 jcm-08-00458-t005:** Prediction of postoperative mortality and occurrence of severe complications basing on preoperative bilirubin concentration.

**Postoperative Mortality**						
**Patients**	**Optimal Cut-Off**	**C-Statistic (95% CI)**	**Sensitivity**	**Specificity**	**Positive Predictive Value**	**Negative Predictive Value**
All patients	≥6.20 mg/dL	0.829 (0.723–0.934)	88.9%	73.2%	29.6%	98.1%
Patients with preoperative albumin concentration ≤3.6 g/dL	≥6.20 mg/dL	0.788 (0.624–0.952)	87.5%	64.3%	41.2%	94.7%
Patients with preoperative hemoglobin concentration ≤12.45 g/dL	≥6.20 mg/dL	0.797 (0.650–0.945)	88.9%	71.0%	47.1%	95.7%
Patients with preoperative biliary drainage	≥6.20 mg/dL	0.891 (0.800–0.982)	87.5%	86.0%	50.0%	97.7%
Patients after right or extended right hemihepatectomy	≥2.73 mg/dL	0.933 (0.853–1.000)	100.0%	78.6%	57.1%	100.0%
**Postoperative Severe Complications**						
**Patients**	**Optimal Cut-Off**	**C-Statistic (95% CI)**	**Sensitivity**	**Specificity**	**Positive Predictive Value**	**Negative Predictive Value**
All patients	≥2.48 mg/dL	0.662 (0.526–0.799)	73.9%	59.6%	42.5%	85.0%
Patients with preoperative creatinine concentration ≥0.77 mg/dL	≥6.20 mg/dL	0.724 (0.559–0.888)	64.7%	84.0%	73.3%	72.8%
Patients with preoperative urea concentration ≥28 mg/dL	≥6.20 mg/dL	0.741 (0.576–0.906)	57.1%	85.2%	66.7%	79.3%
Patients with preoperative biliary drainage	≥2.48 mg/dL	0.685 (0.530–0.840)	70.6%	68.3%	48.0%	84.8%
Patients after right or extended right hemihepatectomy	≥2.48 mg/dL	0.764 (0.585–0.944)	76.9%	78.3%	66.7%	85.7%

95% CI: 95% confidence intervals.

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
