# Peer review of "Relevance of Preoperative Hyperbilirubinemia in Patients Undergoing Hepatobiliary Resection for Hilar Cholangiocarcinoma"

_jcm, 2019, doi:10.3390/jcm8040458_

Reviewer 1 Report

Dear editor

The authors performed a retrospective study with the aim to characterize the associations between the preoperative bilirubin concentration and the risk of postoperative mortality and severe complications on 81 patients who underwent liver and bile duct resection for hilar cholangiocarcinoma. The study, in general, is conducted well, well-written and supported by tables and images. Nevertheless, some methodological limitations remain (retrospective design, small number of patients etc), the results show a week predictive potential and a limited translational value, and, above all, the study lacks originality (the negative effects of high preoperative bilirubin levels on postoperative outcomes are well-known as partially described by the authors). 

Author Response

Response to Reviewer 1 Comments

Dear Reviewer 1,

Thank you for your comments and suggestions.

Point 1: ‘some methodological limitations remain (retrospective design, small number of patients etc)’

Response 1:

The presented study was characterized by all of the limitations that are associated with retrospective research, which was claimed in limitations section in the discussion. However, the analysis covers several years of experience of one of the major centers of hepatobiliary surgery in Europe, and other retrospective studies presenting the experience of individual centers are based on groups of similar numbers, mainly because of low incidence of extrahepatic cholangiocarcinoma. At the same time, low number of patients limited the results and may contribute to omission of other significant relationships. Therefore, I completed the limitation section with the sentence:

‘Small number of patients included in the analysis may lead to exclusion of other significant relationships.’.

Point 2: ‘the results show a week predictive potential and a limited translational value, and, above all, the study lacks originality (the negative effects of high preoperative bilirubin levels on postoperative outcomes are well-known as partially described by the authors).’

Response 2:

Although the negative effects of high preoperative bilirubin concentration is well-known, its influence in case of liver and bile duct resection for extrahepatic cholangiocarcinoma associated with especially high rate of morbidity and mortality is not widely investigated. Similar studies had been performed before, but no consensus has been established yet. Therefore the presented study adds more data on this issue.

Point 3:

English language and style are fine/minor spell check required

Response 3:

I revised the manuscript and corrected spelling and punctuation errors. The changes are highlighted, using the "Track Changes" function in Microsoft Word.

Yours faithfully,

Karolina M. Wronka, M.D., Ph.D.

Department of Hepatology and Internal Medicine, Medical University of Warsaw, 1A Banacha Street, 02-097 Warsaw, Poland

Tel: +48 22 599 16 62

Fax: +48 22 599 16 63

Reviewer 2 Report

There are controversial between the increase in known risk factors raised in issues 1 and 2 and the risk factor relating to hyperbilirubinemia claimed by the authors. Although there is two increased risk factors (well-known) in the "Patients with preoperative biliary drainage" group, all negative situations are judged by only the risk factors of bilirubin increase in the "Patients without preoperative biliary drainage" group.

Issue 1. The determine of clinical prognosis by GGTP activity is well known theory, therefore the worse prognosis would be predicted by increase in the GGTP level. According to the baseline characteristics presented by the authors in Table 1, the GGTP levels of "patients with preoperative biliary drainage" group are significantly increased. It is doubtful whether the clinical relevance between preoperative hyperbilirubinemia and GGTP activity should be considered.

Issue 2. Only 4 patients were judged for TNM tumor staging III or higher in the group of "Patients without preoperative biliary drainage". The number of patients with a lower TNM tumor stage, it can be shown the better prognosis in cancer patients.

Author Response

Response to Reviewer 2 Comments

Dear Reviewer 2,

Thank you for your comments and suggestions.

Point 1:

Are the results clearly presented? - Must be improved.

Response 1:

As far as results are concerned, I modified Table 5 entitled ‘Prediction of postoperative mortality and occurrence of severe complications basing on preoperative bilirubin concentration.’ by adding another column with c-statistic value accompanied with 95% Confidence Interval (95%CI). Thanks to this I limited the number of figures presenting ROC. I excluded figures presenting Receiver operating characteristics curves for prediction of postoperative mortality and occurrence of postoperative severe complications based on preoperative bilirubin concentration among subgroups of patients. After revision the study includes two figures:

‘Figure 1. Receiver operating characteristics curves for prediction of postoperative mortality based on preoperative bilirubin concentration.

Figure 2. Receiver operating characteristics curves for prediction of occurrence of postoperative severe complications based on preoperative bilirubin concentration.’

Point 2: Are the conclusions supported by the results? – Must be improved

Response 2:

With regard to your suggestion to improve the conclusion section, I changed the sentence: ‘This strategy may achieve better early postoperative outcomes regarding the rate of postoperative in-hospital mortality and severe complications after liver and bile duct resection in patients with hilar cholangiocarcinoma.’ into:

‘The results indicate that preoperative biliary drainage should be performed in all patients with hilar cholangiocarcinoma with a bilirubin concentration ≥ 6.00 mg/dL, whereas it should be considered in patients with a bilirubin concentration < 6.00 mg/dL and ≥ 2.50 mg/dL, especially in patients with preoperative hypoalbuminemia, anemia, or renal dysfunction, those who are scheduled to undergo right or extended right hemihepatectomy, and those with ineffective biliary drainage. Preoperative biliary drainage is not recommended in patients with a preoperative bilirubin concentration < 2.50 mg/dL.’

Point 3: ‘Issue 1. The determine of clinical prognosis by GGTP activity is well known theory, therefore the worse prognosis would be predicted by increase in the GGTP level. According to the baseline characteristics presented by the authors in Table 1, the GGTP levels of "patients with preoperative biliary drainage" group are significantly increased. It is doubtful whether the clinical relevance between preoperative hyperbilirubinemia and GGTP activity should be considered.’

Response 3

According to the analysis, gamma-glutamyl transpeptidase activity was significantly higher in patients who underwent preoperative biliary drainage, which might have been associated with intervention within the biliary tree. Probably, there is no or little relevance of elevated gamma-glutamyl transpeptidase activity among patients with preoperative biliary drainage. However, preoperative bilirubin concentration was a risk factor of severe postoperative complications and mortality in this group of patients, therefore bilirubin concentration should be considered even among patients already with preoperative biliary drainage.

In view of your comment, I completed the discussion with:

‘Probably, there is no or little relevance of elevated gamma-glutamyl transpeptidase activity among patients with preoperative biliary drainage. However, preoperative bilirubin concentration was a risk factor of severe postoperative complications and mortality in this group of patients, therefore bilirubin concentration should be considered even among patients already with preoperative biliary drainage.’

Point 4: ‘Issue 2. Only 4 patients were judged for TNM tumor staging III or higher in the group of "Patients without preoperative biliary drainage". The number of patients with a lower TNM tumor stage, it can be shown the better prognosis in cancer patients.’

Response 4:

In the studied group 14 patients with preoperative biliary drainage and 4 patients without preoperative biliary drainage were characterized with TNM tumor staging III or higher. The analysis included only patients qualified for potentially radical liver and bile duct resection. The small number of patients with advanced TNM stage may result from unresectability of advanced hilar cholangiocarcinoma. In numoerous studies the influence of high TNM stage on early outcomes is not investigated. At the same time, lower AJCC T stage has been shown to have positive impact on long-term outcome, including 5-year survival [20, 31] and disease free survival [32]. However, long-term outcomes were not the subject of presented study.

I added new paragraph in the discussion and 2 new references.

‘TNM stage III or higher did not prove to be a significant risk factor for established endpoints. In the studied group 14 patients with preoperative biliary drainage and 4 patients without preoperative biliary drainage were characterized with TNM tumor staging III or higher. The analysis included only patients qualified for potentially radical liver and bile duct resection. The small number of patients with advanced TNM stage may result from unresectability of advanced hilar cholangiocarcinoma. In numerous studies the influence of high TNM stage on early outcomes is not investigated. At the same time, lower AJCC T stage has been shown to have positive impact on long-term outcome, including 5-year survival [20, 31] and disease free survival [32]. However, long-term outcomes were not the subject of presented study.’

‘31. Buettner, S.; Margonis, G.A.; Kim, Y.; Gani, F.; Ethun, C.G.; Poultsides, G.; Tran, T.; Idrees, K.; Isom, C.A.; Fields, R.C.; Krasnick, B.; Weber, S.M.; Salem, A.; Martin, R.C.; Scoggins, C.R.; Shen, P.; Mogal, H.D.; Schmidt, C.; Beal, E.; Hatzaras, I,; Shenoy, R.; Maithel, S.K.; Pawlik, T.M. Conditional probability of long-term survival after resection of hilar cholangiocarcinoma. HPB (Oxford). 2016, 18, 510–517. DOI: 10.1016/j.hpb.2016.04.001.

32. Hu, H.J.; Mao, H.; Shrestha, A.; Tan, Y.Q.; Ma, W.J.; Yang, Q.; Wang, J.K.; Cheng; N.S.; Li, F.Y. Prognostic factors and long-term outcomes of hilar cholangiocarcinoma: A single-institution experience in China. World J Gastroenterol. 2016, 22, 2601–2610. DOI: 10.3748/wjg.v22.i8.2601.’

Point 5:

English language and style are fine/minor spell check required

Response 5:

I revised the manuscript and corrected spelling and punctuation errors. The changes are highlighted, using the "Track Changes" function in Microsoft Word.

Yours faithfully,

Karolina M. Wronka, M.D., Ph.D.

Department of Hepatology and Internal Medicine, Medical University of Warsaw, 1A Banacha Street, 02-097 Warsaw, Poland

Tel: +48 22 599 16 62

Fax: +48 22 599 16 63

 Round  2

Reviewer 1 Report

The Authors added some clarifications in the text, and corrected spelling errors. They also improved sections of the paper, especially the conclusions, which are now clearer. The translational value of the findings is now better evidenced.

Reviewer 2 Report

All concerns that raised by this reviewer have been well addressed.